# Multiscale Feature Extraction by Using Convolutional Neural Network: Extraction of Objects from Multiresolution Images of Urban Areas

Ching-Lung Fan [ID]

Department of Civil Engineering, Republic of China Military Academy, Kaohsiung 83059, Taiwan;
p93228001@ntu.edu.tw; Tel.: +886-7-456290

**Abstract:** The emergence of deep learning-based classification methods has led to considerable advancements and remarkable performance in image recognition. This study introduces the Multiscale Feature Convolutional Neural Network (MSFCNN) for the extraction of complex urban land cover data, with a specific emphasis on buildings and roads. MSFCNN is employed to extract multiscale features from three distinct image types—Unmanned Aerial Vehicle (UAV) images, high-resolution satellite images (HR), and low-resolution satellite images (LR)—all collected within the Fengshan District of Kaohsiung, Taiwan. The model in this study demonstrated remarkable accuracy in classifying two key land cover categories. Its success in extracting multiscale features from different image resolutions. In the case of UAV images, MSFCNN achieved an accuracy rate of 91.67%, with a Producer's Accuracy (PA) of 93.33% and a User's Accuracy (UA) of 90.0%. Similarly, the model exhibited strong performance with HR images, yielding accuracy, PA, and UA values of 92.5%, 93.33%, and 91.67%, respectively. These results closely align with those obtained for LR imagery, which achieved respective accuracy rates of 93.33%, 95.0%, and 91.67%. Overall, the MSFCNN excels in the classification of both UAV and satellite images, showcasing its versatility and robustness across various data sources. The model is well suited for the task of updating cartographic data related to urban buildings and roads.

**Keywords:** multiscale feature extraction; convolutional neural network; unmanned aerial vehicle image; satellite image

## 1. Introduction

Remote sensing involves the use of cameras and sensors installed on aerospace-borne platforms for directly observing large areas of the Earth's surface to enable the timely and cost-effective mapping of land cover patterns. The urban information that can be extracted from remote-sensing images can contribute to urban planning, transportation network updates, land cover mapping, and other applications. Furthermore, urban land use maps have considerable value to those wishing to monitor, plan, and design urban environments [1]. Buildings and roads are the most common artificial structures in urban areas, and the automated extraction of data on surface structures is crucial for urban land use mapping. Urban land use maps are currently updated through aerial photograph interpretation and field surveys, which are labor-intensive and time-consuming tasks. Advancements in remote-sensing technology have enabled the acquisition of a large number of high-spatial-resolution (HSR) remote-sensing images, including images covering urban areas. These images are obtained using sensors installed on satellites or unmanned aerial vehicles (UAVs). Satellites and UAVs can be employed to acquire an abundance of data, such as photographs, three-dimensional (3D) point clouds, and videos, which can be analyzed using algorithms and software [2]. The classification of aerial and satellite imagery plays a crucial role in numerous applications, such as land cover and land use mapping, change detection, and emergency response and management [3].

Satellites and aircraft (e.g., manned aerial vehicles and UAVs) offer different advantages to those wishing to acquire ground imagery through remote sensing. Satellites can obtain extensive ground imagery, and their performance is less affected by weather conditions. Very-high-resolution (VHR) satellite imagery, which can now be obtained because of advancements in remote-sensing technology, has been used for the classification of land features in many studies. For example, Gibril et al. [4] employed WorldView-2 imagery to map asbestos cement roofs in urban areas. Huang et al. [5] used WorldView imagery to classify land in Hong Kong and Shenzhen, China, into 11 land use categories. Borana and Yadav [6] employed high-resolution (HR) satellite data for land-use suitability mapping and land-use land cover (LULC) analysis in the urban area. Aircraft and UAVs are capable of maneuvering over and rapidly collecting ground imagery on small areas, thereby enabling the acquisition of highly detailed imagery at very fine spatial resolution (VFSR) in complex urban areas. VFSR aerial imagery has been used for classifying land cover and land use in urban areas. For example, Zhang et al. [7] analyzed 10 major land cover categories by using VFSR aerial imagery of the cities of Southampton and Manchester in the United Kingdom. Researchers have employed UAV-obtained HSR imagery of urban areas in various applications, such as the automated detection of damaged stone pavements [8] and urban vegetation mapping [9].

Because of the complexity of land features in urban areas, researchers encounter challenges when extracting urban land cover information from HSR imagery. Zhao et al. [10] reported that extracting land use information from HSR remote-sensing imagery is beneficial. Single land parcels with a specific purpose (e.g., residential, commercial, or industrial) often encompass multiple land cover types with different spatial, spectral, and geometric characteristics. For example, a residential area might contain trees, buildings, and water bodies, which complicates the automated mapping of land use. In addition, urban areas of similar land use types, such as residential areas, often have unique physical attributes or employ unique land cover materials (e.g., roofs made from different tile types), whereas areas with different land use types might have similar or overlapping reflectance spectra and textures (e.g., asphalt roads and parking lots) [11]. Furthermore, the urban land features that can be obtained from VFSR imagery are highly complex and diverse, often containing combinations of artificial urban and semi-natural surfaces in close proximity [12]. Majd et al. [13] reported variation in object proportions within VHR imagery, which further complicates scene classification. Therefore, land use classification is challenging because of the complexity and diversity of spatial and structural patterns in urban areas [14].

Remote-sensing technology is a powerful tool for collecting ground surface information and has become the primary means of generating large-scale land cover datasets [15]. Various research methods have been employed to extract information on artificial structures in urban areas from diverse types of remote-sensing data. Buildings and road information can be extracted from satellite imagery [16], aerial imagery [17], UAV imagery [18], lidar data [19], synthetic aperture radar imagery [20], and hyperspectral imagery [21]. These remote-sensing data possess high spatial and temporal resolution. However, the primary challenge associated with the utilization of remote-sensing data is their accurate and effective classification [22]. Advancements in remote-sensing technology have facilitated the acquisition of data with high spatial and spectral resolutions. Machine learning has emerged as the most frequently used technique for classifying remote-sensing images. The selection of the most appropriate classification algorithm is a common topic in research involving images with different spatial and spectral resolutions captured from various platforms.

Traditional machine learning methods employed for remote-sensing data classification include maximum likelihood estimation [23] and cluster analysis [24,25]. More advanced techniques such as support vector machines [26], random forests [27,28], artificial neural networks (ANNs) [29], and convolutional neural networks (CNNs) [30] are also being increasingly utilized. The initial algorithms for land cover classification were ANNs, which primarily perform supervised or unsupervised classification by statistically analyzing

image data [31]. ANNs have found widespread application in land use and land cover classification [32]. However, noise in data can reduce the accuracy of image classification using ANNs, and their training can be time consuming [33]. CNNs were developed as supervised learning neural networks to overcome the limitations of ANNs. Due to their deep architecture, CNNs offer advantages over traditional ANNs in image recognition and classification tasks.

CNN-based deep learning methods are increasingly being used in applications related to land cover and land use, thereby enabling various types of land information to be updated. Deep learning is a crucial branch of machine learning in which multilayer neural networks are used to construct models from features. Considerable advancements in deep learning have been made in fields such as facial recognition, autonomous driving, and natural language processing. CNNs can efficiently retrieve complex patterns and informative features from satellite imagery, and they outperform support vector machines and random forests in such retrieval [34]. Moreover, CNNs outperform traditional ANNs in image recognition and classification tasks. The computational complexity involved in implementing CNNs is lower than that involved in implementing other types of networks, and CNNs require fewer weight values. In addition, CNNs can directly process images as inputs, thereby enabling automatic feature extraction without the requirement of manual engineering. Because of their high computational efficiency and accuracy, CNNs can perform multiclass image recognition, object detection, and land cover classification.

In recent years, deep learning methods utilizing CNNs have seen increasing use in land cover classification. For instance, Giang et al. [35] employed the U-Net framework to develop a land cover prediction model, utilizing multispectral UAV imagery for training. Zhang et al. [36] proposed a classification method based on the enhanced DeepLabv3+ network and optimized the classification results for land cover using a fully connected conditional random field (CRF). Behera et al. [37] introduced a multiscale CNN framework for semantic segmentation in urban land cover and conducted experiments using two image datasets: (1) the NITRDrone dataset and (2) the urban drone dataset (UDD).

Extracting semantic features from complex scenes, especially scenes of urban areas, is challenging because of the heterogeneity, considerable intraclass variations, and small interclass variations of such scenes [18]. Additionally, the extraction of spectral features from images is influenced by the radiometric characteristics of the images, where lighting conditions play a crucial role. Consequently, a given object can exhibit various features under different shadows or lighting conditions, and different types of objects might have similar spectral characteristics under visible light. CNNs can account for radiometric variation between neighboring pixels and have high flexibility for capturing relevant shapes in images rather than relying solely on objects' color (or radiometric) features. Furthermore, because satellite imagery often lacks high spectral resolution, differentiating between object classes solely based on spectral information is challenging. Therefore, the class to which pixels belong can be inferred based on the context and shapes of surrounding objects. CNNs can automatically learn hierarchical contextual features from input images, and their architecture is well suited to extracting information from VHR imagery [38]. The effectiveness of CNNs for classifying VHR imagery and extracting specific objects has been well established [39].

Road networks are widely used in various applications and thus are essential sources of information [40]. In low-resolution and medium-resolution satellite imagery, roads appear as thin and long structures. In high-resolution and VHR satellite imagery, roads often appear as long homogeneous regions with certain widths [41]. This variation in the same object across images with different resolutions means that effective classification methods are required. Traditional machine learning methods based on pixel- or object-level analysis are unsuitable for classifying land cover in images with various resolutions. Moreover, radiation noise originating from vehicles, ground markings, and shadows causes high spectral variability within a class [42]. Variations in the color and texture of buildings, particularly large buildings, in high-resolution imagery, might lead to incomplete or partial

extraction results [43]. Extracting data on roads and buildings in urban areas is challenging because of similarities in the spectral reflectance and texture properties of these two types of structures [44].

Numerous studies have investigated the extraction of building or road data from remote-sensing imagery using various convolutional neural network (CNN) architectures. For instance, Zhang et al. [45] employed a fully convolutional network to extract building data. Majd et al. [13] proposed an object-based deep CNN framework for extracting information about different types of buildings. They tested their approach on aerial imagery of the Vaihingen area in Germany and satellite imagery (WorldView-2) of the Tunisia region. Manandhar et al. [46] developed a general model for road classification by integrating a CNN with volunteered geographic information (VGI). They tested their model on satellite imagery from Abu Dhabi, United Arab Emirates, and aerial imagery from Massachusetts, United States. Similarly, Chen et al. [47] used CNN and VGI models to identify buildings in Malawi and roads in Guinea. Younis et al. [48] employed a neural network called SegNet, which has an encoder–decoder architecture, for the segmentation of buildings and roads in images.

In CNNs, various convolutional kernel (i.e., filter) sizes can be used to extract multiscale features. By using multiscale features in image classification, CNNs can capture information on various details in an image, such as the shape and edges of small objects, the parts and overall shape of medium-sized objects, and the background of large objects. Combining features from various scales can enhance the performance and robustness of models. In a multiscale feature extraction block, channel-wise weights are applied to every channel of the multiscale feature; this process results in the emphasis on features that are beneficial for classification and the suppression of irrelevant features [49]. Therefore, Sun et al. [50] proposed multiscale convolutional neural networks, which achieve accurate building extraction at various scales by leveraging multiscale deep features, employing an SVM-based decision fusion strategy, and optimizing results with superpixels, resulting in reduced noise and enhanced structural integrity in building extraction.

This paper introduces a deep learning model that integrates multiscale feature extraction techniques to improve feature identification for classification purposes. Specifically, we developed a CNN that incorporates multiscale feature extraction to efficiently identify buildings and roads in images. The focus of this research is on the application of the Multiscale Feature CNN (MSFCNN) method, which plays a pivotal role in image classification across various resolutions. Therefore, the MSFCNN method is applied with refined hyperparameter tuning to classify images obtained at different resolutions. The primary objective of this study is to address the classification challenges arising from variations in the representations of identical objects across diverse resolutions. The model is intricately designed to adeptly extract data related to buildings and roads from images featuring varying resolutions, such as UAV images, high-resolution satellite imagery, and low-resolution satellite imagery. Its application across diverse image resolutions underscores its utility in extracting pertinent information for the refinement and updating of urban land use maps. This strategic approach ensures the provision of reliable information crucial for updating urban land use maps.

## 2. Research Method

A CNN is a deep learning network that learns hierarchical representations from training data. The weights of CNNs can be adjusted and shared on the basis of training data, their performance can be generalized and optimized, their parameters can be simplified, and they can automatically reduce the number of parameters with high feature discrimination and extraction capabilities [51]. This section describes the main architecture, operation, and functionalities of CNNs.

## 2.1. CNNs

A CNN is composed of one or multiple convolutional layers, pooling layers, and fully connected layers, which—through feature learning—collectively generate detection results for images or objects. A CNN incorporates convolutional and pooling layers, which differentiates it from a traditional ANN. CNNs have machine vision abilities for perceiving fine details in images and do not solely rely on calculations based on extracted data, as is the case with other types of neural networks. During the CNN training process, features sensitive to specific classes are automatically detected. This process occurs through local connections and weight sharing, which reduces the number of trainable parameters in the neural network and enhances the learning efficiency. These characteristics enable CNNs to learn image features automatically, and they contribute to the widespread use of CNNs in applications such as image recognition, classification, detection, and segmentation. The typical structure, functions, and hyperparameters of a conventional CNN are described in the following text.

### 2.1.1. Convolutional Layer

Convolution is achieved by sliding a specified-size window, which is also known as a filter or kernel, over an input image. This sliding process occurs sequentially from left to right and from top to bottom, with the filter typically having the same stride in the height and width directions. The region covered by the filter is referred to as the receptive field. In the convolutional layer, a dot product (i.e., matrix multiplication) is obtained by multiplying the values of the filter matrix by the pixel values of the image. The resulting products are then summed, and, finally, a bias value is added to this sum, as presented in Equation (1). A large sum of dot products indicates that a certain shape in the input image closely matches the shape of the filter. As the filter slides over all positions of the image, a matrix of values, which is known as the feature map, is generated (Figure 1). Multiple feature maps can be present in a convolutional layer, thereby enabling the model to learn different feature functions.

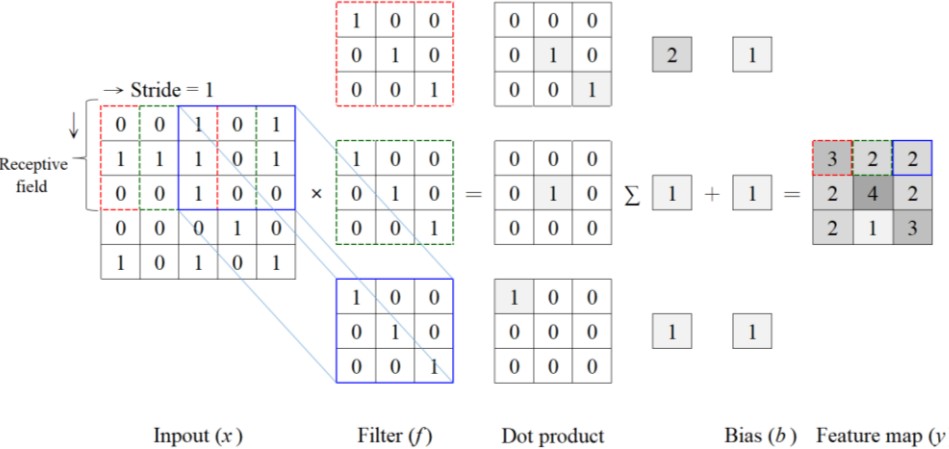

**Figure 1.** The feature calculation process of the convolutional layer.

$$y = \sum (x_{ij} \times f_{ij}) + b \tag{1}$$

A nonlinear activation function is commonly applied after each convolutional layer. This step is crucial for introducing nonlinear features into the model because the convolutional layer primarily performs linear computations, such as element-wise multiplication and summation. ReLU is often employed as a nonlinear activation function because it avoids the problem of vanishing gradients and ensures high training speed without compromising accuracy. The ReLU function, which is represented in Equation (2), takes the

input $x$ of the ReLU activation layer and produces the output $y$. The gradient of ReLU is either 0 or 1.

$$\text{ReLU} = \max(0, x)$$
$$\frac{dy}{dx} = \begin{cases} 1, & x \geq 0 \\ 0, & x < 0 \end{cases} \tag{2}$$

### 2.1.2. Multiscale Feature Extraction

Objects in images vary in size and shape, and features extracted at a single scale might not effectively capture all relevant information on objects shown in images. Multiscale feature extraction involves using multiple convolutional kernels or pooling layers with different scales to capture the features of objects at various scales. This technique enables a classifier to identify objects more accurately, especially when images show objects of various sizes.

Multiscale feature extraction is an effective technique for enhancing the accuracy and robustness of classifiers and provides excellent performance for the recognition of objects of varying sizes within images. CNNs use the receptive field to perceive features at different scales. If the receptive field is too small, the network can only capture local features; if the receptive field is too large, excessive noise might be captured. By leveraging convolutional operations, CNNs can adjust the receptive field's size and extract multiscale information. Consequently, these networks help to prevent information loss and effectively capture features at various scales. Obtaining multiscale features from an input image requires convolution with kernels of various sizes. The input feature map is convolved using these multiple kernels; for each output, the network learns the feature map of a single convolutional operator. The calculation of the feature map is expressed in Equation (3), in which $F$ denotes the output feature map, $n$ denotes the number of channels in the output feature map, $K$ denotes the convolutional kernel, and $kl$ and $kw$ denote the height and width of the convolutional kernel, respectively.

$$F_{l', w', n} = \sum_{kl, kw, m} K_{kl, kw, m, n} \cdot X_{l' + kl - 1, w' + kw - 1, m} \tag{3}$$

Multiscale feature extraction is a technique employed in image processing and computer vision that aims to capture and represent information at different spatial scales within an image. The fundamental principle behind multiscale feature extraction is to analyze an image at multiple resolutions or scales, extracting relevant features at each level. This process enables the model to perceive and understand both fine details and coarse structures within the image, contributing to a more comprehensive representation. The process involves decomposing the image into multiple scales, extracting relevant features independently at each scale, and then combining these features to create a comprehensive representation. In the first step, the image undergoes pyramid decomposition, generating different scales. Subsequently, features are extracted at each scale, capturing fine details and coarse structures. The extracted features from various scales are then concatenated or fused to form a unified set of features. In the context of deep learning, these features contribute to learning hierarchical representations within the neural network, enhancing the model's ability to generalize and recognize patterns at different levels of granularity.

### 2.1.3. Pooling Layer

The pooling layer, which is also referred to as the subsampling layer, plays a crucial role in downsampling the spatial dimensions of the feature map through the application of a predefined function (e.g., maximum or average pooling) over local regions. This process effectively reduces the size of the input image and preserves its red–green–blue (RGB) depth. The main objectives of the pooling layer are to retain essential features from the feature map, reduce computational complexity, and prevent overfitting.

Three main methods are used in pooling layers: average pooling, maximum pooling, and stochastic pooling. The choice of pooling method depends on the requirements of

the specific task. Average pooling is an effective technique for reducing variance and preserving background information in images, thereby mitigating errors caused in the feature extraction process by the size of the neighborhood in the image. Maximum pooling is effective when errors occur in the estimation of the convolutional layer's parameters; this process reduces the effect of such errors and highlights features such as textures. The function of stochastic pooling is intermediate between those of average pooling and maximum pooling. Stochastic pooling involves assigning probabilities to the magnitudes of elements and implementing sampling on the basis of these probabilities. Figure 2 depicts a 2 × 2 filter with a stride of 1, where the average value of the receptive field is employed in the downsampling operation.

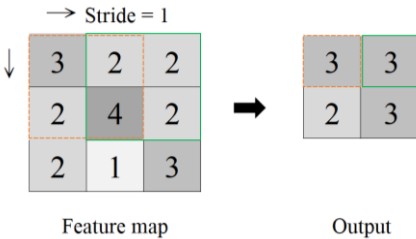

**Figure 2.** Average pooling with pool size of 2 × 2 and stride of 1.

### 2.1.4. Fully Connected Layer

The primary function of the fully connected layer is to learn the associations between features and their corresponding classes by assigning specific weights. In this layer, the dot product between these weights and the preceding layer is calculated, which enables accurate probabilities to be obtained for different classifications. In a convolutional layer, each feature map represents a specific feature of the input signal, and as the number of layers increases, the features become more abstract. A fully connected layer is used to integrate these convolutional features and produce the final output. A probability or label is then predicted using an activation function, such as the softmax function or sigmoid function, on the output. Thus, the model can make predictions and classify the input on the basis of the features obtained from all convolutional layers.

Through backpropagation, the network adjusts its weights on the basis of the error or loss calculated during forward propagation; the predictions made by the model thus improve over time. This process is essential for training a neural network through gradient descent optimization. The loss function involves the calculation of the error between the predicted and true values; that is, it quantifies the discrepancy between the predicted output and the ground truth. By employing a loss function, the neural network adjusts the weights and biases of its neurons to minimize the error (loss). A loss function commonly used in deep learning is cross entropy, which is expressed in Equation (4), in which $Y_i'$ denotes the encoded true label vector for class $i$ (0 or 1), whereas $Y_i$ denotes the output probability label vector for class $i$.

$$\text{Cross Entropy} = -\sum Y_i' \cdot \log(Y_i) \tag{4}$$

The weights (denoted $W$) are updated by randomly assigning an initial set of weights $W_i$. Through backpropagation, the network identifies the weights that contribute the most to the loss and determines the adjustment method that can be used to minimize the loss. Gradient descent is then employed to calculate the slope ($dL/dW$) of the loss function with respect to the weights. Subsequently, the weights are updated using Equation (5), in which $\eta$ denotes the learning rate, which is the rate at which the weights are updated through gradient descent.

$$W = W_i - \eta \frac{dL}{dW} \tag{5}$$

### 2.1.5. Data Augmentation and Dropout

Overfitting occurs when a model becomes too closely fit to the training samples, which results in poor performance on the validation and test sets. Overfitting is a common long-standing challenge encountered when machine learning algorithms are used. To mitigate this problem, researchers have proposed techniques such as data augmentation and dropout. These techniques are beneficial for CNNs, which often require a large quantity of training data to achieve optimal performance. Data augmentation involves introducing minor variations to the training set, thereby altering the arrays of the training data, and preserving labels. The augmentation process generates additional training images through the application of various types of transformation or modification methods, such as rotation, scaling, flipping, and noise addition. Expanding the training dataset in this manner enhances a model's generalizability and reduces the risk of overfitting.

Dropout is a common technique used to address the problem of overfitting during the training of deep learning models. Dropout involves randomly deactivating a subset of parameters during model training, specifically by disabling certain feature detectors or neurons with a certain probability. This approach ensures that the neural network does not excessively rely on specific training samples. The main function of dropout is to disconnect a certain percentage of hidden units or neurons randomly, thereby reducing the model's reliance on specific local features. The dropout technique provides feature representations that are more generalizable than those obtained without dropout, thereby enabling the creation of a more robust model that is less susceptible to overfitting.

In this study, the MSFCNN operates through three steps, as illustrated in Figure 3. The first step involves image input and data augmentation, starting from the input layer, where images with varying resolutions from satellites or drones are input into the network. Data augmentation techniques are applied to increase the quantity of images. The second step is image feature extraction, utilizing multiple convolutional kernels with different sizes to capture features at various scales. Pooling reduces spatial dimensions, and features are downsampled from different scales. Hyperparameter tuning is performed in this process, recording the performance of each hyperparameter combination and its related training performance to achieve the optimal model configuration for generalization capability. The key focus in multiscale feature extraction is the integration of convolutional layers with various kernel sizes to enhance the model's ability to recognize features of different sizes in the input data. The third step involves transforming the output into a one-dimensional array (flattened layer), followed by fully connected layers to learn high-level representations. The loss function measures the difference between predicted values and actual values, guiding the updating of weights and biases during training. The final output layer utilizes the softmax activation function for classification, producing the ultimate prediction results.

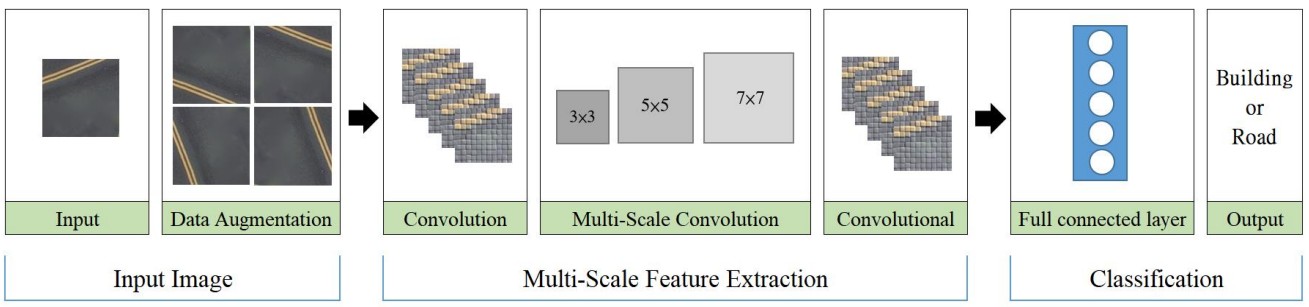

**Figure 3.** Three operational steps of this research method.

### 2.2. Study Area and Image Data

Situated in southern Taiwan, Kaohsiung exhibits a distinctive array of geographical characteristics. Notably, it stands as Taiwan's largest port city, ranking among the world's busiest ports. While an urban metropolis, Kaohsiung is also replete with an abundance of verdant parks and open spaces. Moreover, it encompasses several industrial zones and

manufacturing facilities, constituting a substantial portion of its economic landscape. These geographical attributes collectively contribute to Kaohsiung's multifaceted and dynamic character, harmoniously blending urban development with cultural richness.

In this study, Fengshan District in Kaohsiung, Taiwan, was selected as the research area (Figure 4a). UAV image data were collected using the DJI Phantom-3 UAV equipped with the FC300C camera (focal length of 4 mm). This UAV was flown at an altitude of 60 m. The collected images included RGB channels, and the image resolution was 4000 × 3000 pixels (Figure 4b). Data from the Environmental Systems Research Institute World Imagery Map, which offers satellite (Figure 4c) with a resolution of 1 m or higher from various locations worldwide, were also employed. Within the ArcGIS user community, multiple imagery options are available, such as ultra-high-resolution imaging at a scale of 1:280 (down to 0.03 m). The high-resolution World Imagery utilized in this study is sourced from the WorldView-3 satellite. WorldView-3 operates as a sun-synchronous satellite orbiting at an altitude of 617 km, completing one full orbit around the Earth every 97 min. It provides a comprehensive range of eight spectral bands, including red, green, blue, NIR1, coastal blue, red-edge, yellow, and NIR2, thus offering a diverse multispectral dataset. The multispectral imagery boasts a spatial resolution of 1.2 m, while the panchromatic imagery reaches an impressive spatial resolution of up to 0.3 m.

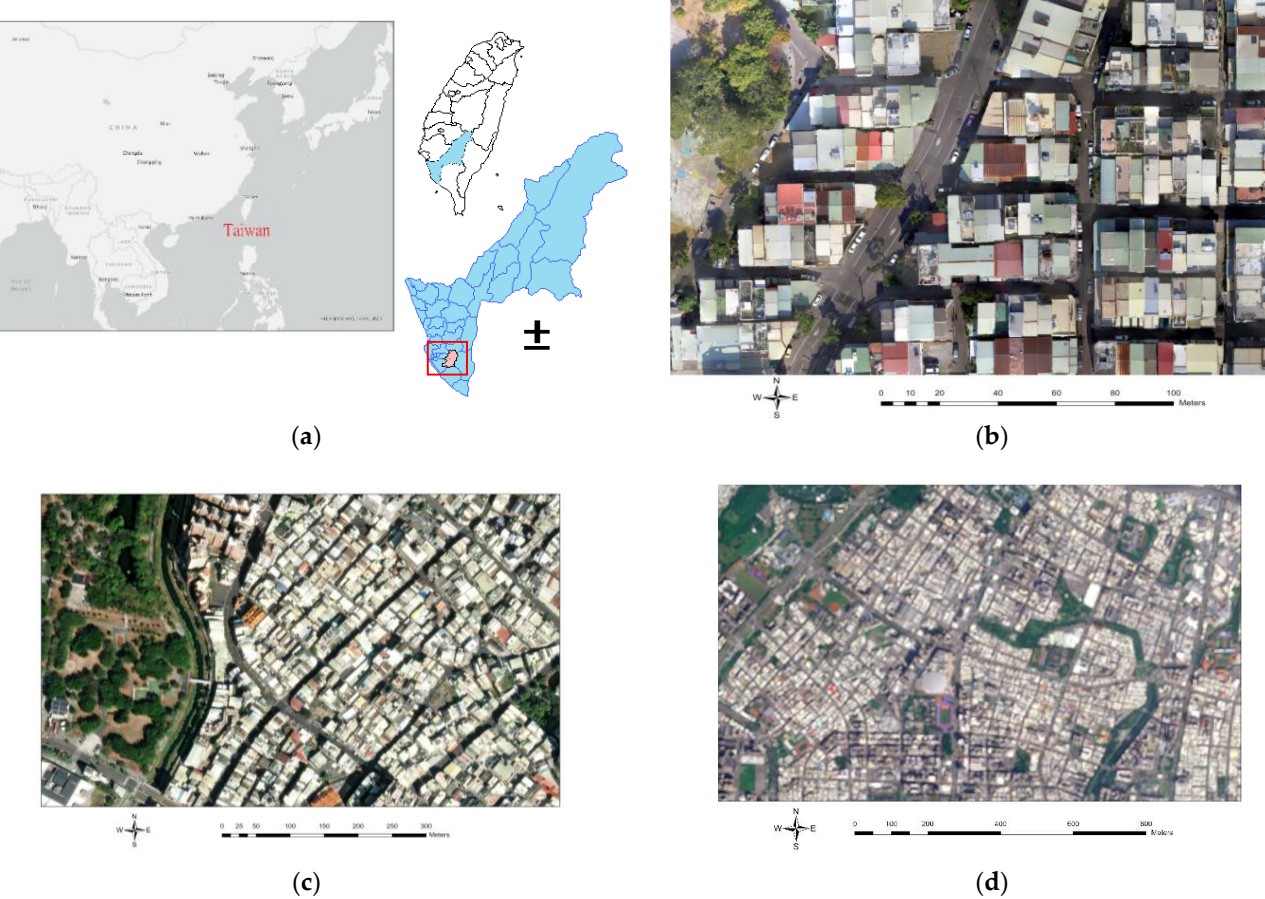

**Figure 4.** The UAV and satellite images of the study area: (**a**) study area; (**b**) UAV images; (**c**) high-resolution satellite images; (**d**) low-resolution satellite images.

Sentinel-2 is a group of Earth observation satellites developed and operated by the European Space Agency (ESA) for the European Union's Copernicus program. These satellites are designed for the purpose of monitoring environmental conditions and changes in land cover on the Earth's surface. Sentinel-2's low-resolution imagery, typically at 10 m per pixel, provides a broader view of the Earth's surface with less fine-grained

detail (Figure 4d). This imagery is valuable for monitoring large-scale environmental changes, land cover classification, and regional assessments. It is particularly useful when a wider perspective is needed, and fine details are less critical. The three types of remote-sensing images used in this study had considerably different resolutions and characteristics; therefore, they were ideal for testing the generalizability of the proposed model in extracting data on buildings and roads in urban areas. The effectiveness of the constructed CNN model was validated using these imagery data with various resolutions.

This study primarily employs the Multiscale Feature CNN (MSFCNN) to extract models for buildings and roads and to train and test on two types of land cover images. To enhance the model's performance while training, the image dataset is divided into training, validation, and testing sets. The training set is primarily used for model training, the validation set for parameter tuning, and the testing set for evaluating the model's performance. Datasets for building and road features are created by cropping UAV images and satellite images. Each category, building, and road, comprises 200 samples. Subsequently, a total of 1200 images from UAV images, high-resolution (HR), and low-resolution (LR) satellite images are randomly split into 600 images for the training set, 240 for the validation set, and 360 for the testing set, maintaining a proportion of 50%, 20%, and 30%, respectively (Figure 5). By employing the designed MSFCNN architecture and utilizing an extensive image dataset under varying image resolution conditions, this approach effectively extracts various land cover features in the real world. This is particularly important for complex and dynamic urban areas, specifically in identifying buildings and roads.

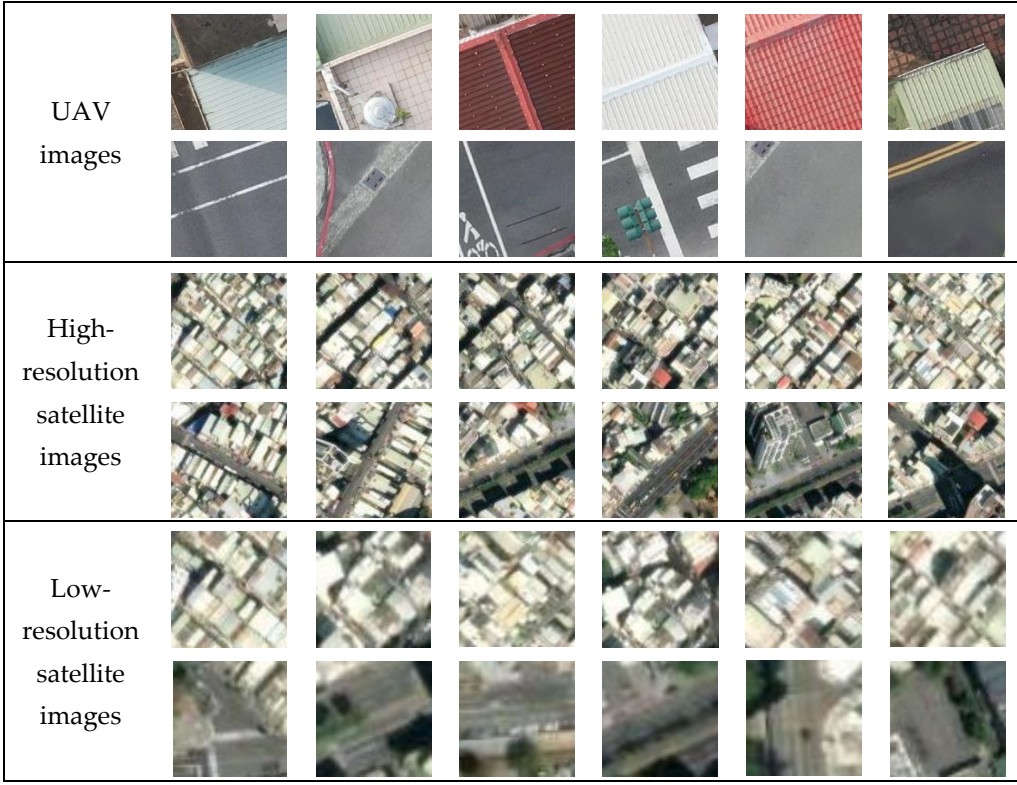

**Figure 5.** Datasets of buildings and roads in UAV images and satellite images.

### 2.3. Image Preprocessing and Model Construction Process

Image preprocessing is crucial to ensure that a CNN can effectively learn and extract meaningful features from input images. These preprocessing steps include (a) image cropping: this initial step involves cropping, focusing on specific regions of interest within the images. (b) Image resizing: images are resized to a consistent dimension, ensuring uniform width and height for all input images. (c) Label encoding: categories or classes

for each image are encoded for training. This encoding enables the model to predict class probabilities for each category (Figure 6).

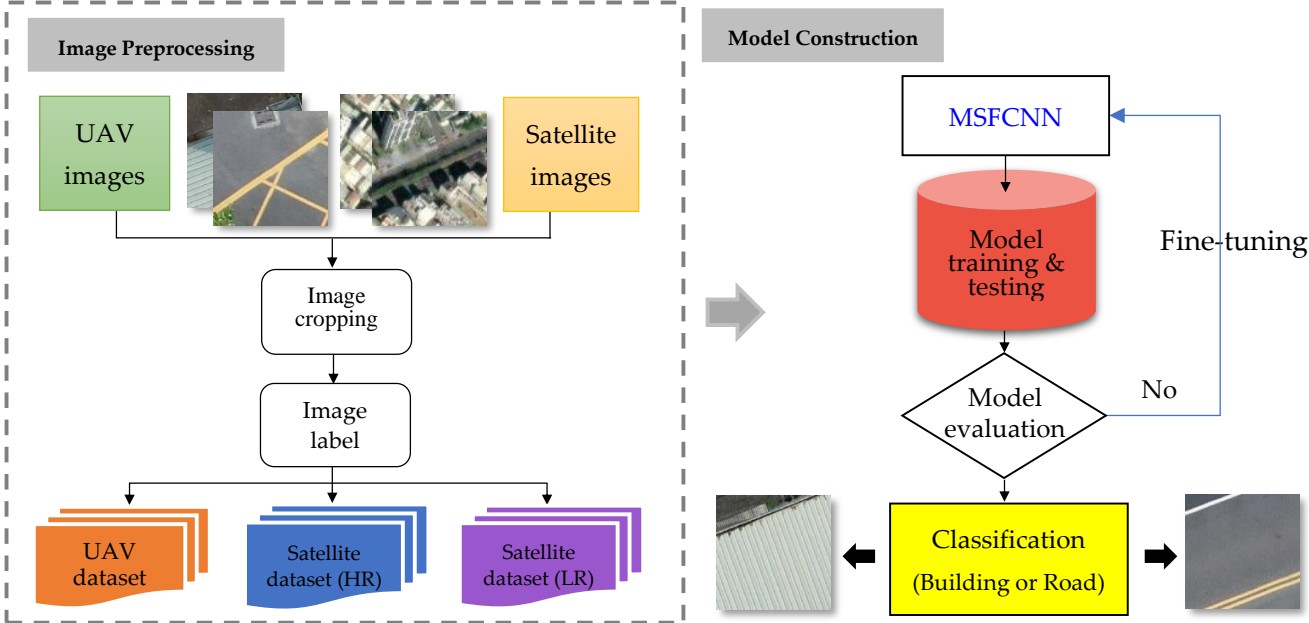

**Figure 6.** The process of image preprocessing and MSFCNN construction in this study.

The construction process of the Multiscale Feature CNN (MSFCNN) model for extracting building and road features involves the following key steps. (a) Model architecture: The MSFCNN architecture is thoughtfully designed to accommodate multiscale feature extraction. It is tailored to handle the diverse and complex characteristics of building and road features in urban settings. (b) Resolution: Given the availability of images with varying resolutions, the MSFCNN framework is strategically configured to effectively process both high-resolution (HR) and low-resolution (LR) images. This adaptation accounts for the different scales at which features need to be detected. (c) Parameter fine-tuning: The model is trained using the training dataset while simultaneously being validated with the validation dataset. This iterative process allows for fine-tuning and optimization of the model's parameters to ensure optimal performance. (d) Model Evaluation: The final evaluation is conducted using the testing dataset to measure the model's ability to accurately extract building and road features. Metrics such as accuracy, Producer's Accuracy, and user's accuracy are employed to quantitatively assess the model's performance.

## 3. Results and Analysis

The deep neural network used in this study was a CNN model with a moderate number of filters in each layer. This design choice was motivated by the fact that the classification task involved only two classes: buildings and roads. In contrast to multiclass classification problems that require large-scale neural networks such as VGG16, the classification problem of this study was a binary classification problem. Furthermore, the adopted neural network model was relatively small and offered advantages in terms of training and prediction time; therefore, this model was suitable for urban mapping and updating tasks. Notably, the proposed MSFCNN exhibited excellent performance and generalizability for UAV and satellite images.

### 3.1. CNN Architecture and Hyperparameters

A CNN is a supervised learning deep neural network, and its architecture plays a crucial role in determining its computational efficiency and classification accuracy. The number of layers, the number and size of filters in each layer, and other structural aspects

considerably affect model performance. Understanding the roles and principles of the network layers and parameters is crucial for constructing deep neural network architectures that satisfy practical requirements. The MSFCNN structure used in this study comprised one input layer, three convolutional layers, three pooling layers, and one fully connected layer (including a flattening layer, a hidden layer, and an output layer), as illustrated in Figure 7. In this study, Python served as the primary tool for conducting the research. The Python programming language was employed during both the training and testing phases of the CNN model. In addition, the Keras library, a high-level API that leverages deep learning libraries like TensorFlow, was utilized for the feature extraction process. The study involved the utilization of Python for implementing the CNN-multiscale feature extraction model. Additionally, the authors devised a Python script to facilitate the conversion of two resolution scenarios into input images suitable for the CNN model, as well as to translate the model's output into representations of buildings and roads.

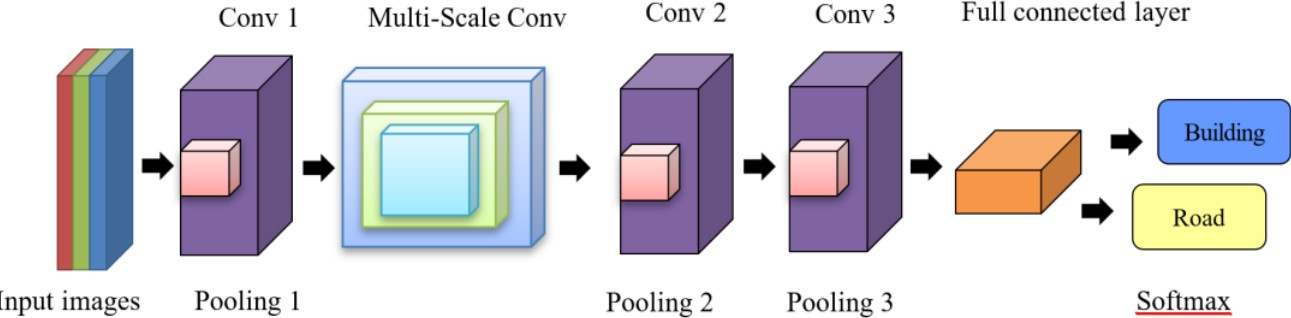

**Figure 7.** Design of MSFCNN architecture.

The MSFCNN conducted depth-wise separable convolution operations on input feature maps. In these operations, sets of convolution operators with different sizes were used to extract feature information at various scales. This approach enabled the modeling of contextual information, captured global dependencies, and reduced computational costs. First, the input layer comprised samples with a resolution of $150 \times 150 \times 3$ pixels, where the dimensions represent the image height, image width, and channel (RGB), respectively. Second, the convolutional layers had a stride of 1 and a padding of 1. The stride refers to the step size at which the filter slides over the spatial input. A larger stride results in a smaller output, which potentially causes the loss of some input features, but lower computational costs. The Conv1 layer used 64 filters with a size of $3 \times 3$. Conv2 and Conv3 contained 128 and 256 filters, respectively (i.e., the number of filters was doubled from one layer to the next). In the multiscale convolution, filters with sizes of $3 \times 3$, $5 \times 5$, and $7 \times 7$ were used, with 128 filters for each size. The ReLU activation function was applied to the outputs generated from all the convolutional layers and the inputs generated from the fully connected layers. Third, regarding the pooling layer, a pooling size of $2 \times 2$ was applied, and maximum pooling was employed for all calculations. To mitigate overfitting, a dropout rate of 0.5 was employed; thus, 50% of the neurons in the neural network were randomly disabled during each training iteration. Fourth, the fully connected layer comprised a flattening layer\, two hidden layers with neuron sizes of 1000 and 2000, respectively, and two output layers, which provided the classification results (i.e., building or road). Finally, after the convolution, pooling, and multiscale feature extraction processes, the softmax function was applied to implement the binary classification of the building or road in each input image. Table 1 provides a summary of the hyperparameter settings for each layer of the constructed CNN.

**Table 1.** MSFCNN architecture and hyperparameters.

| Item | Input Layer | Convolutional Layer | Pooling Layer | Fully Connected Layer |
|---|---|---|---|---|
| 1 | | Conv 1: 64 filters = 3 × 3, stride = 1, padding = 1, activation = ReLU | Pool 1: Max pooling = 2 × 2, dropout = 0.5 | |
| 2 | 400 images: 150 × 150 × 3, rescale = 1/255, rotation range = 40, width shift range = 0.2, height shift range = 0.2, shear range = 0.2, zoom range = 0.2, horizontal flip = true | Multiscale Conv: 128, filters = 3 × 3, stride = 1, padding = 1, filters = 5 × 5, stride = 1, padding = 2, filters = 7 × 7, stride = 1, padding = 3, activation = ReLU | | Input: activation = ReLU Hidden 1: 1000 neuron Hidden 2: 2000 neuron Output: activation = softmax |
| 3 | | Conv 2: 128 filters = 3 × 3, stride = 1, padding = 1, activation = ReLU | Pool 2: Max pooling = 2 × 2, dropout = 0.5 | |
| 4 | | Conv 3: 256 filters = 3 × 3, stride = 1, padding = 1, activation = ReLU | Pool 3: Max pooling = 2 × 2, dropout = 0.5 | |

The MSFCNN model developed in this study employs filters with sizes of 3 × 3, 5 × 5, and 7 × 7 for multiscale feature extraction. This strategic use of varying filter sizes enables the aforementioned model to capture feature information at multiple scales, thereby facilitating the modeling of contextual information and global dependencies, in addition to substantially reducing computational costs. An innovative aspect of the developed MSFCNN model is that it performs depth-wise separable convolution operations. These operations involve employing sets of convolution operators with different sizes to capture features effectively across various scales. This innovative approach contextualizes information and exhibits robust performance for images with different resolutions. The MSFCNN model is particularly advantageous for urban mapping tasks involving images captured from UAV and satellite platforms; its adaptability to different scales renders it a versatile and efficient tool for extracting meaningful features in urban environments.

### 3.2. Model Training, Validation, and Testing

A loss value is produced during each epoch of a training process. This value indicates the error between the predicted classifications of a CNN model and the actual class labels. Training continues until a predetermined number of epochs has been reached, after which the weights in the network are updated using algorithms such as stochastic optimization or batch gradient descent. For the training set, training is continued until the errors are minimized or a stopping criterion is reached. Validation has two purposes. First, it prevents model overfitting by ensuring that the model does not rely solely on the training set data. Second, it aids in hyperparameter tuning. Hyperparameters are crucial settings in CNNs and are manually adjusted and configured to optimize a model's output. Examples of hyperparameters are learning rate, number of epochs, number of hidden layers, activation functions, and batch size. A test set, the data of which are not used during training or validation, is used to validate the performance of a CNN model and ensure its generalizability. A CNN model is evaluated using a test set once it has generated satisfactory results for a training set and validation set. A well-designed and randomly selected test set is crucial for accurately evaluating the performance of a CNN model on diverse real-world image features.

The dataset used in this study comprised 400 images, with 200 images each of buildings and roads obtained from UAV and satellite platforms. These images were cropped and annotated. To achieve optimal feature extraction by using the constructed MSFCNN model, this model was trained on batches of 32 images per iteration during stochastic gradient descent. The weights were updated during the training process by using randomly selected batches of images. One complete pass through the entire dataset was considered an epoch. For each epoch, the MSFCNN was trained using 200 images from the training set, validated using 80 images from the validation set, and tested using 120 images from the testing set. The training process spanned 30 epochs, and the aim was to achieve accurate classification or feature extraction for the training data.

When the model was trained on the UAV image dataset for 30 epochs, it achieved an accuracy of 92.11% (Figure 8a). Furthermore, the model achieved an accuracy of 91.98% on the testing set. The CNN learning process has similarities with general supervised learning, with both processes requiring a large quantity of labeled data for training. However, CNNs have the unique ability to adjust weights iteratively through repeated training on the same samples; this adjustment minimizes the discrepancy between the predicted and actual labels and ultimately enables accurate classification. The learning loss on the training data was 0.23 in epoch 10 and then improved to the optimal value of 0.21 in epoch 30 (Figure 8b). Experiments were also performed on high-resolution satellite images (HR), with the number of images and training epochs being the same as those used for the UAV image dataset. The accuracy of the model trained on the satellite images gradually converged to 88.28% in epoch 10 and finally reached 92.22% (Figure 9a). The CNN model achieved an accuracy of 92.12% on the testing set. In epoch 30, the learning loss of the training model was 0.19 (Figure 9b). In the MSFCNN-trained model, the low-resolution satellite images (LR) achieved an accuracy of 94.21% (Figure 10a). The model also demonstrated an accuracy of 92.76% on the testing set. By epoch 30, the training model had reached a learning loss of 0.22 (Figure 10b). It is noteworthy that the accuracy and loss of the MSFCNN model on the training set exhibit minimal deviation from the results obtained with high-resolution (HR) images.

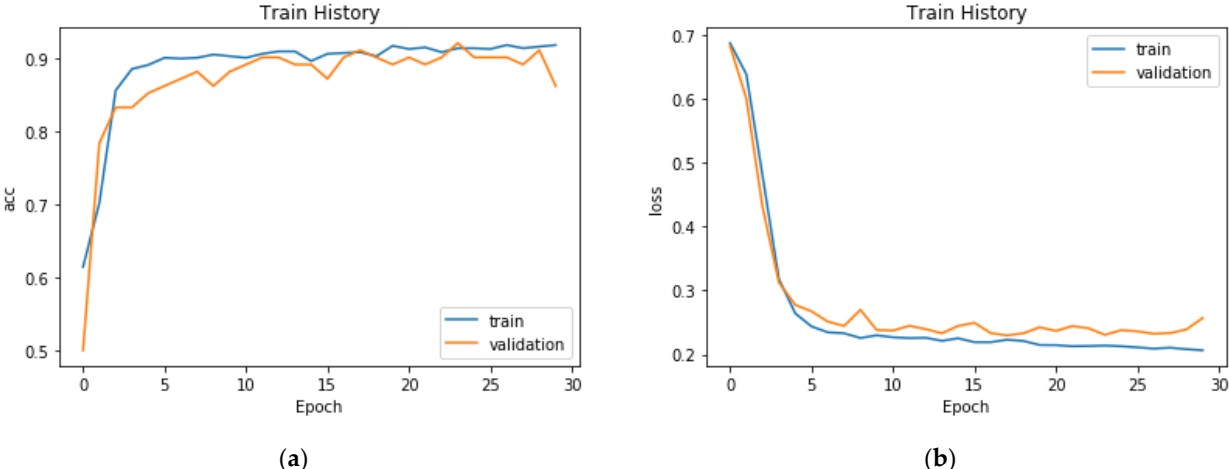

(**a**)        (**b**)

**Figure 8.** The MSFCNN on training and validation set for UAV images: (**a**) model's accuracy; (**b**) model's loss.

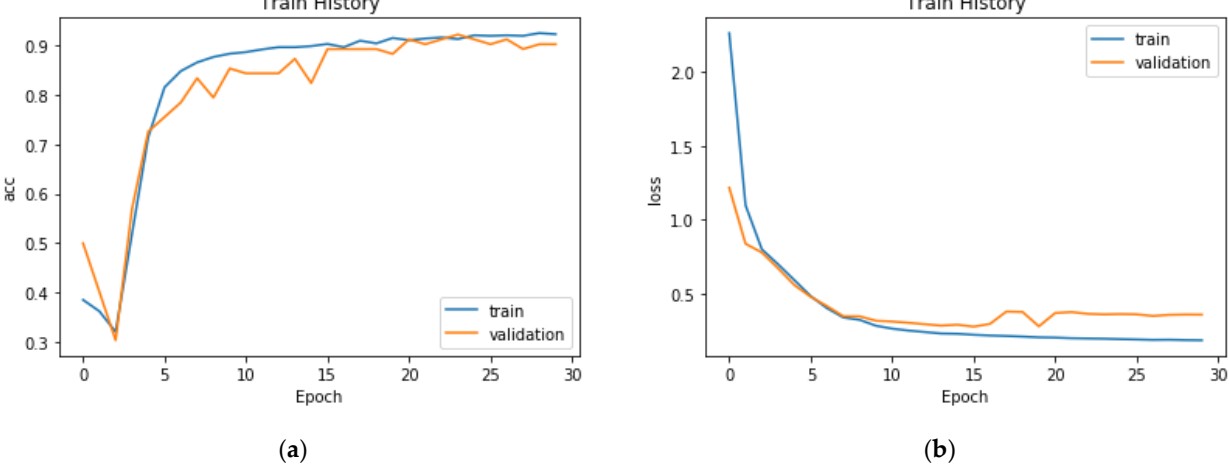

(**a**)        (**b**)

**Figure 9.** The MSFCNN on training and validation set for satellite images (HR): (**a**) model's accuracy; (**b**) model's loss.

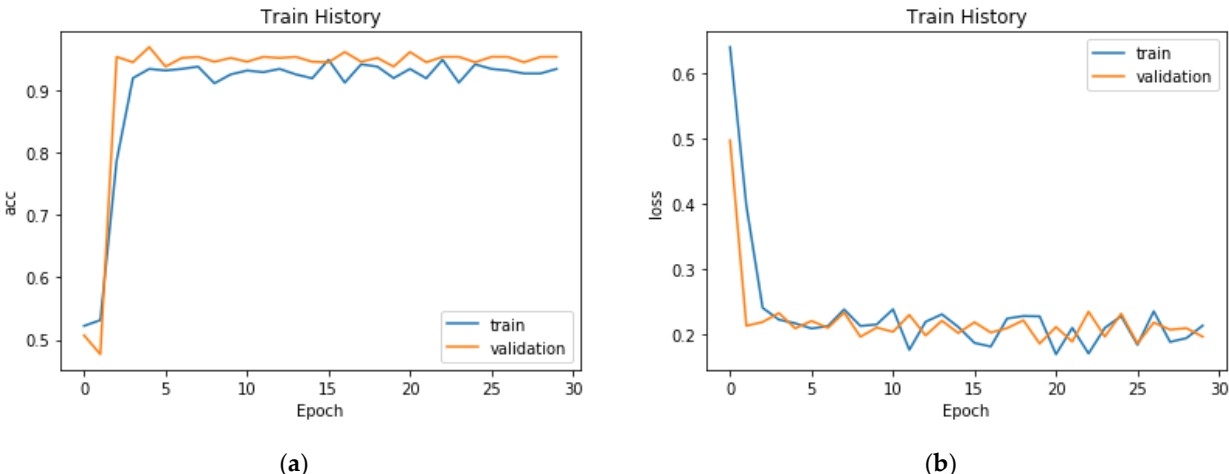

(**a**)                                                    (**b**)

**Figure 10.** The MSFCNN on training and validation set for satellite images (RL): (a) model's accuracy; (b) model's loss.

This study performed optimization training and testing for the constructed MSFCNN's structure and hyperparameters. The input layer can process images with a resolution of $150 \times 150 \times 3$ pixels, and the Conv1, Conv2, and Conv3 layers use 64, 128, and 256 filters with a size of $3 \times 3$, respectively. The incremental doubling of filters enhances the MSFCNN model's capacity to capture complex and abstract features. This model uses filters with sizes of $3 \times 3$, $5 \times 5$, and $7 \times 7$ to capture feature information at various scales. The ReLU activation function is executed after each convolutional operation on the inputs of the fully connected layers. The nonlinearity introduced by this function increases the expressive power of the model, thereby enabling the model to learn complex input-to-output mapping. The fully connected layer includes two hidden layers: one containing 1000 neurons and the other containing 2000 neurons. The learning rate and number of epochs are set to 0.01 and 30, respectively, to facilitate the task of feature learning through backpropagation and the classification of urban land cover classes by using the softmax function. The depth-wise separable convolution operations of the proposed model not only enhance its performance but also optimize its computational efficiency, thereby rendering it suitable for real-world applications involving resource constraints.

The developed MSFCNN model exhibited excellent performance in various aspects, irrespective of whether UAV or satellite imagery was used (Table 2). This model accurately extracted data on buildings and roads from different image sources. Notably, the MSFCNN's learning error and overfitting were at low levels, and the model achieved an accuracy of approximately 92% on the test sets for UAV and satellite imagery. The CNN model's excellent extraction performance is attributable to its ability to incorporate the image features of both buildings and roads. By leveraging a large quantity of training data, the proposed MSFCNN model can effectively extract object features within a certain range, thereby addressing the challenges encountered in object classification. Moreover, the proposed CNN model can simultaneously extract buildings and roads from images with different resolutions, which further underscores its versatility and effectiveness.

Producer's Accuracy (PA) is a statistical measure commonly used in remote-sensing, classification, and accuracy assessment of classification models, particularly in the context of image analysis and geographic information systems (GIS). It evaluates the ability of a classification model or algorithm to correctly identify and classify a specific class. PA is typically defined and calculated: True Positives/(True Positives + False Negatives). Here, True Positives (TP) represent the number of correctly classified buildings and roads. These are the samples that the model correctly identified as belonging to the target class. False Negatives (FN) represent the number of instances of the class of interest that were incorrectly classified as a different class or category by the model. These are the samples that the model failed to detect or missed.

**Table 2.** Performance results of MSFCNN.

| Image Type | Accuracy of the Training Set (%) | Accuracy of the Validation Set (%) | Accuracy of the Test Set (%) | Learning Error | Overfitting |
|---|---|---|---|---|---|
| UAV images | 92.11 | 86.27 | 91.67 | 0.21 | 0.13 |
| Satellite images (HR) | 92.22 | 90.20 | 92.50 | 0.19 | 0.10 |
| Satellite images (LR) | 94.21 | 95.31 | 93.33 | 0.04 | 0.01 |

User's Accuracy (UA) assesses the probability that a specific class assigned by a classification model to a dataset sample indeed belongs to that class. UA is typically defined and calculated: True Positives/(True Positives + False Positives). UA is particularly important in situations where false positives (misclassifying non-members of a class as members) can have significant consequences. UA helps evaluate the precision or reliability of the classification model when it assigns a particular class label. It answers the question, "When the model predicts a certain class, how often is it correct?" A high UA value suggests that when the model predicts a specific class, it is usually correct, which indicates a reliable model.

Table 3 presents the confusion matrix for MSFCNN classification of UAV images, with an accuracy of 91.67%. It indicates that out of 120 images of buildings and roads, 110 were correctly classified. The PA stands at 93.33%, with 60 true instances of buildings, of which 56 were correctly classified as buildings, and 4 were misclassified as roads. The UA is 90.0%, signifying that out of the 60 actual buildings, 6 were erroneously classified as roads. Similarly, Table 4 presents the confusion matrix for MSFCNN classification applied to high-resolution satellite images (HR), yielding accuracy, PA, and UA values of 92.5%, 93.33%, and 91.67%, respectively. These metrics align with the corresponding values obtained for low-resolution satellite images (LR), which are 93.33%, 95.0%, and 91.67%, as shown in Table 5.

**Table 3.** Performance of the MSFCNN using the test set in UAV images.

| Ground Truth | Classification Results | | Producer's Accuracy, PA (%) |
|---|---|---|---|
| | Building | Road | |
| Building | 56 | 4 | 93.33 |
| Road | 6 | 54 | 90.0 |
| User's Accuracy, UA (%) | 90.0 | 93.33 | Accuracy = 91.67% |

**Table 4.** Performance of the MSFCNN using the test set in high-resolution satellite images.

| Ground Truth | Classification Results | | Producer's Accuracy, PA (%) |
|---|---|---|---|
| | Building | Road | |
| Building | 56 | 4 | 93.33 |
| Road | 5 | 55 | 91.67 |
| User's Accuracy, UA (%) | 91.67 | 93.33 | Accuracy = 92.50% |

**Table 5.** Performance of the MSFCNN using the test set in low-resolution satellite images.

| Ground Truth | Classification Results | | Producer's Accuracy, PA (%) |
|---|---|---|---|
| | Building | Road | |
| Building | 57 | 3 | 95.0 |
| Road | 5 | 55 | 91.67 |
| User's Accuracy, UA (%) | 95.0 | 91.67 | Accuracy = 93.33% |

The slightly higher accuracy of MSFCNN in extracting buildings and roads from low-resolution imagery compared to high-resolution imagery can be attributed to several factors:

(a) Low-resolution imagery typically demands less precision in handling noise and fine details, rendering MSFCNN more robust when operating at lower resolutions. High-resolution images often contain more fine details and noise, which might introduce additional interference into the model.

(b) In low-resolution images, large-scale features, such as the overall outlines of buildings or roads, are generally easier to recognize. MSFCNN may excel in processing these large-scale features, whereas high-resolution images could contain more small-scale details requiring more intricate processing.

(c) High-resolution images may carry more noise or imperfect data, potentially affecting model performance adversely. In contrast, low-resolution images usually exhibit greater tolerance to these issues.

It is important to note that there was little difference in accuracy between satellite imagery and UAV imagery in this study. Nevertheless, the model's performance across various resolutions may vary depending on the application, dataset, and model architecture. Enhancing performance on high-resolution images might necessitate more data and a more complex model architecture.

## 4. Discussions

### 4.1. Comparison of Other Neural Networks

To compare the performance of MSFCNN with other CNNs in land cover land use (LCLU) classification, we conducted a review of six papers published between 2020 and 2023. Due to a lack of detailed information regarding the network architectures and hyperparameters in the majority of these articles, our analysis was limited to descriptions of different models' LCLU categories, image data, and their respective performance metrics. Consequently, we were unable to perform an in-depth comparison on the same dataset. The models under consideration encompass the encoder–decoder framework (EDF) [52], deep learning (DL) [53], Multilayer Perceptron Neural Network (MLPNN) [54], shared and specific feature learning (S2FL) [55], pyramid feature extraction (PFE) [56], and multimodal bilinear fusion network (MBFNet) [57]. Quantitative assessment was performed using three widely used metrics in image classification and segmentation, namely accuracy, precision (PA), and recall (UA).

Li et al. [52] proposed an EDF CNN that integrates spatial-aware circular modules and a semantic distribution alignment loss function for LU classification. The model achieved an accuracy of 84.2% on optical and synthetic aperture radar (SAR) imagery in Ezhou and Panjin, China. Abdi [53] compared the performance of SVM, RF, Xgboost, and DL for Sentinel-2 LU in south-central Sweden. The results showed that SVM led in accuracy (75.8%), followed by Xgboost (75.1%), RF (73.9%), and DL (73.3%). Girma [54] employed the MLPNN (Artificial Neural Network with Cellular Automata–Markov Chain) to model LULC change in the Gidabo River Basin, Main Ethiopian Rift, from 1985 to 2050 using satellite imagery. The MLPNN achieved an overall accuracy of 85.9~87.04%. Hong et al. [55] proposed an S2FL model for processing multimodal remote-sensing data. Three benchmark datasets—Houston2013, Berlin, Augsburg—were used for LC classification. Experimental results show that the accuracy of the S2FL model in these datasets is 85.07%, 62.23%, and 83.36%, respectively. Li et al. [56] introduced a PFE for building extraction in high-resolution remote-sensing images. The proposed method utilizes attention modules and a structural-cue-guided feature alignment module to address challenges in single-scale depth features. Applied to the WHU Building Dataset, the method achieves an F1 score of 95.3% and an IoU score of 90.9%. Li [57] proposed MBFNet with Second-order Attention-based Channel Selection for LC classification using optical and SAR images. Three coregistered optical and SAR datasets—PoDelta (Italy), ChongMing (Yunnan, China), and WuHan (Hubei, China)—were employed. Experimental results show that the accuracy of the MBFNet model in these datasets is 93.61%, 82.61%, and 78.22%, respectively.

Table 6 displays the classification results for the seven models on various images, providing a comprehensive context for evaluating the performance of the MSFCNN model and the other models. In terms of accuracy, the MSFCNN method consistently maintains the highest values among all the models. Overall, the MSFCNN proposed in this study has demonstrated superior performance across all metrics, affirming the effectiveness of this approach. Surprisingly, the DL model outperforms other methods in the UA evaluation metric, including MLPNN and MSFCNN, both using Sentinel-2 imagery, but the differences between the latter two are not substantial. The developed MSFCNN model is suitable for extracting buildings and roads in urban areas. This model outperformed other models in LULC classification tasks, particularly in terms of accuracy, demonstrating its effectiveness in suppressing false positives. Compared with that of the MLPNN model, which is similar to the MSFCNN model, the average accuracy of the MSFCNN model was higher by 6.29% (93.33% vs. 87.04%). The MSFCNN model consistently outperformed the other models in LULC classification, confirming its robustness and the effectiveness of the proposed approach. Although the developed model exhibited excellent UA results, these results should be interpreted with caution because this model is a specialized tool for extracting buildings and roads in urban areas for LULC classification.

**Table 6.** Comparison results of different neural networks in land cover and land use.

| Method | LCLU | Image | Accuracy (%) | PA (%) | UA (%) |
|---|---|---|---|---|---|
| EDF [52] | Seven LU | Optical and SAR image | 84.2 | NA | NA |
| DL [53] | Eight LU | Sentinel-2 | 73.3 | 36~95 | 63~98 |
| MLPNN [54] | Nine LCLU | Landsat-5, Landsat-7, and Sentinel-2 | 85.9~87.04 | 75.9~96.6 | 73.3~96.7 |
| S2FL [55] | Seven LC | Hyperspectral, and SAR image | 62.23~85.07 | NA | NA |
| PFE [56] | Building | Aerial images | NA | 93.4~96.0 | 92.9~95.3 |
| MBFNet [57] | Five LC | Optical and SAR images | 78.22~93.61 | 74.4~92.5 | 75.5~86.4 |
| MSFCNN | Building and road | UAV, WorldView-3, and Sentinel-2 | 91.67~93.33 | 90.0~95.0 | 90.0~95.0 |

Since the performance levels of the models compared in this study were obtained on different datasets, caution should be exercised in interpreting these performance results. Despite the inherent challenges associated with cross-dataset comparisons, the results of this study indicate the adaptability and effectiveness of the MSFCNN model across diverse LULC scenarios involving different image resolutions, thus affirming its robustness and applicability to urban LULC classification tasks.

## 4.2. The Constraints and Novelties

The constraints and challenges of using multiscale feature extraction through CNN in this study are as follows: (a) The utilization of CNNs for multiscale feature extraction demands substantial computational resources, especially when applied to large-scale and high-resolution image datasets. High-performance hardware (GPU) is essential, which can pose challenges for some research projects or applications. (b) Training MSFCNN effectively requires a vast amount of labeled data that comprehensively covers various scenarios and features. The acquisition and preparation of such annotated datasets can be time consuming and labor intensive, making it a significant challenge, particularly for specific domains or applications with limited data availability. (c) CNNs are susceptible to overfitting, especially when the training dataset is insufficient or lacks diversity. Overfitting can lead to a reduction in model performance when applied to new and unseen data. (d) Multiscale Data Processing: Handling data from multiple scales and resolutions necessitates intricate preprocessing steps to ensure that the model effectively captures features at different scales. (e) Optimizing the performance of MSFCNN models requires

tuning numerous hyperparameters, including learning rates, layer configurations, and the number of neurons. Finding the optimal parameter combinations can entail a significant amount of experimentation and computational resources. Addressing these constraints and challenges necessitates meticulous problem analysis, adept data preprocessing techniques, model fine-tuning, and efficient utilization of computational resources.

This research introduces a deep learning model named multiscale feature convolutional neural network (MSFCNN) for multiresolution urban land cover classification. The key novelties of this model are:

(1) Multiscale convolutional architecture

The integration of various convolutional filter sizes ($3 \times 3$, $5 \times 5$, $7 \times 7$) within a single CNN model provides it with the unique capability to perceive both fine details and global structures in the imagery. This facilitates the simultaneous capture of low-level details as well as high-level semantic information, allowing the model to perceive objects effectively across scales. While previous works have explored multiscale concepts, this paper is novel in employing it for the specific use case of classifying essential urban land cover types (buildings, roads) from multiresolution aerial/satellite images. The design choices targeting this application demonstrate the viability of multiscale convolutional neural networks for automated urban mapping tasks.

(2) Cross-resolution robustness

A rigorous evaluation protocol assessing the model's accuracy across UAV images as well as high-resolution and low-resolution satellite images establishes its consistency irrespective of input image resolution. The model achieves consistent performance across inputs with widely varying resolution levels. This versatility to perform well with both high-detail and coarse-resolution imagery underscores the robustness of the proposed approach. Benchmarking against multiple image types is innovative and provides convincing evidence of the model's generalizability, addressing a key gap in the existing literature on deep neural networks for remote-sensing applications.

(3) Domain-centric optimization:

The model parameters are deliberately tuned based on the traits of buildings and roads in urban regions through rigorous hyperparameter optimization. This specialized optimization enhances model performance. The model architecture and hyperparameters are purposely optimized for extracting buildings and roads through a series of tweaks—filter sizes, the number of feature maps, and training strategies. This specific domain enables superior performance on the target application compared to general-purpose networks. Very few studies have investigated optimized deep CNNs for this niche yet important application of automating the mapping of key urban land cover categories. The proposed innovations thus address an unmet need.

In summary, the integration of multiscale convolution to perceive both fine and coarse features, paired with customization for classifying essential urban land cover categories, enables the proposed MSFCNN model to deliver state-of-the-art results across diverse image resolutions. This demonstrates its viability as an automated tool for updating urban land use maps.

## 5. Conclusions

The complexity of images of urban areas makes it challenging to classify these images into different classes by using traditional classification methods and limits the ability of these methods to extract features from images of buildings and roads. In addition, these methods often fail to achieve consistent and satisfactory classification results when they are applied to images with various resolutions. To address these challenges, this paper proposed a multiscale feature convolutional neural network (MSFCNN) for extracting data on buildings and roads from images with three resolutions. MSFCNN, which has multiple filters in its convolutional layers, effectively overcomes the challenge of feature extraction by determining the optimal representations of images. The MSFCNN model constructed in this study is tailored to the size and geometric shape of objects. Highly

accurate outputs were achieved for the classification of objects in urban areas through the thoughtful design of the CNN architecture, hyperparameter tuning, rigorous training, and comprehensive testing.

This study leveraged the MSFCNN to achieve remarkable results in the extraction of buildings and roads. MSFCNN demonstrated a high degree of accuracy in classifying two primary land cover categories, regardless of image resolution. In the case of UAV images, it achieved an impressive accuracy rate of 91.67%. Similarly, when applied to high-resolution satellite images (HR) and low-resolution satellite images (LR), they achieved respective accuracy rates of 92.5% and 93.33%, respectively.

Despite the significant successes of the MSFCNN model, it is crucial to acknowledge its limitations. The effectiveness of MSFCNN can be further evaluated with additional datasets and diverse urban environments to ensure its generalizability. Additionally, the model's performance may vary when confronted with specific challenges, such as varying weather conditions, inconsistent lighting, or the presence of occlusions. For future research, it is recommended to explore the adaptation of MSFCNN to address the challenges associated with varying environmental conditions. The research will focus on developing image processing and analysis systems with enhanced computational capabilities with respect to spectral characteristics and texture analysis. Moreover, the inclusion of additional sensor data, such as LiDAR or thermal imaging, could enhance the model's performance and further broaden its applications in urban land cover classification. Lastly, conducting comparative studies with other state-of-the-art models may provide valuable insights into the strengths and weaknesses of MSFCNN in different contexts.

**Funding:** This research received no external funding.

**Data Availability Statement:** Data are contained within the article.

**Conflicts of Interest:** The author declares no conflict of interest.

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
