# Peer review of "Multiscale Feature Extraction by Using Convolutional Neural Network: Extraction of Objects from Multiresolution Images of Urban Areas"

_ijgi, doi:10.3390/ijgi13010005_

Round 1
Reviewer 1 Report (Previous Reviewer 1)
Comments and Suggestions for Authors
Dear Authors,
Thank you for taking in to account my suggestions, however, i still think you need to report the output classified maps from your classification, from each datasets in the result section. Please do so.
Regards.
Author Response
Thank you for your valuable feedback and suggestions. We appreciate your suggestion to report the output classified maps from each dataset in the results section. However, it is important to note that the employed Multi-Scale Feature Convolutional Neural Network (MSFCNN) is primarily designed for classification tasks, such as distinguishing between buildings and roads, and does not inherently support semantic segmentation for object-wise delineation. Despite the inherent limitations of the MSFCNN architecture in generating classified maps for each dataset as requested, we would like to highlight its excellent performance in the context of classification tasks for multiresolution images.
Due to the nature of the network architecture, generating classified maps in the traditional sense, which represent segmented objects, is not within the capability of the MSFCNN architecture. While our methodology may not provide the specific classified maps as suggested, we believe that the detailed classification results we can provide for each dataset offer valuable insights into the performance of our approach. We are grateful for your understanding of the limitations of our method and appreciate your guidance.
Additionally, we recognize the importance of further advancements in the field, and although MSFCNN may not currently support semantic segmentation, we are actively exploring potential future improvements and modifications to enhance its capabilities. Once again, we sincerely thank you for your constructive feedback and assistance throughout this review process.
Reviewer 2 Report (Previous Reviewer 3)
Comments and Suggestions for Authors
The topic is interesting. However, the novelty is limited.
1) Research method is naive and simple, the proposed approach lacks the novelty.
2) The experiments are not convincing. It is difficult to validate the advantages of the proposed approach from Table 2-5. Table 6 is unfair since they are based on different datasets.
Comments on the Quality of English LanguageEnglish should be improved by the native speaker.
Author Response
Dear reviewer,
We would like to express our sincere gratitude to the reviewer for their tremendous help throughout the review process. We appreciate the reviewer's time, patience, and constructive criticisms, all of which have been immensely valuable to us. In response to the reviewers' insightful comments, we have made several modifications to the paper to address their concerns. Additionally, we have provided a detailed point-by-point response to each of the reviewers' comments. To facilitate easy identification, the modifications have been highlighted in blue within the manuscript.

Reviewer 3 Report (New Reviewer)
Comments and Suggestions for Authors
The paper presents a Multi-Scale Feature Convolutional Neural Network (MSFCNN) model for extracting complex urban land cover data. The model is designed for extracting data on buildings and roads from images with various resolutions: Unmanned Aerial Vehicle (UAV) images, high-resolution satellite images (HR), and low-resolution satellite images (LR).
- The manuscript is a well-written, well-organized and well-illustrated.
- The literature review is well structured and provides relevant background information.
- The results of this work are important as they leveraged the MSFCNN to achieve the extraction of buildings and roads from images.
- Author should explicitly demonstrate the contribution of the approach. The author should present works that based on MSFCNN, and demonstrate the contribution of the proposed approach with regards to these works.
Author Response
Dear reviewer,
We would like to express our sincere gratitude to the reviewer for their tremendous help throughout the review process. We appreciate the reviewer's time, patience, and constructive criticisms, all of which have been immensely valuable to us. In response to the reviewers' insightful comments, we have made several modifications to the paper to address their concerns. Additionally, we have provided a detailed point-by-point response to each of the reviewers' comments. To facilitate easy identification, the modifications have been highlighted in blue within the manuscript.

Reviewer 4 Report (New Reviewer)
Comments and Suggestions for Authors This paper highlights the novelty of the MSFCNN model, emphasizing its application for extracting multiscale features from UAV, high-resolution satellite, and low-resolution satellite images. The contribution of the study is clearly stated, indicating the remarkable accuracy achieved in classifying key land cover categories, especially buildings and roads. However, before publication, addressing the following questions is imperative.One primary concern is the clarity regarding whether the MSFCNN method is a novel proposal or merely an application, a distinction not evident in the initial three sections of the paper. The Literature Review section currently primarily addresses research progress, and if this is the case, consolidating it with the Introduction may enhance coherence. If maintained as a standalone section, it should pivot towards a more comprehensive summary and analysis of research progress, expressing the author's perspectives on the strengths and limitations of different research methods. Additionally, the Research Method section should provide a more conspicuous description of the MSFCNN method.
A second noteworthy point is the absence of comparative experimental studies in the paper. While the author acknowledges the challenges in replicating experiments due to the lack of specific information and offers a preliminary descriptive comparative analysis in the discussion section, a paper focusing on method application must include comparative experiments for a more comprehensive assessment.
Line 53: It notes the country, and a similar notation should be made at line 68 for Hong Kong and Shenzhen, China.
Line 355: Please include the country information as "China."
Author Response
Dear reviewer,
We would like to express our sincere gratitude to the reviewer for their tremendous help throughout the review process. We appreciate the reviewer's time, patience, and constructive criticisms, all of which have been immensely valuable to us. In response to the reviewers' insightful comments, we have made several modifications to the paper to address their concerns. Additionally, we have provided a detailed point-by-point response to each of the reviewers' comments. To facilitate easy identification, the modifications have been highlighted in blue within the manuscript.

Round 2
Reviewer 2 Report (Previous Reviewer 3)
Comments and Suggestions for Authors
The novelty needs being clarified and enhanced.
Author Response
Response:
We appreciate your feedback and constructive comments. We acknowledge the point you raised regarding the limited clarity and enhancement of the novelty in our paper. In response to this concern, we have revised the manuscript, providing a more detailed and explicit explanation of the novelty of our approach. By offering a clearer exposition of the novelty, we aim to better convey the significance and original contributions of our work. We have taken care to elaborate on the distinctive features that set our model apart from existing approaches and highlight how these contributions advance the field.
This research introduces a deep learning model named multi-scale feature convolutional neural network (MSFCNN) for multiresolution urban land cover classification. The key novelties of this model are:
(1) Multiscale convolutional architecture:
The integration of various convolutional filter sizes (3x3, 5x5, 7x7) within a single CNN model provides it with the unique capability to perceive both fine details and global structures in the imagery. This facilitates the simultaneous capture of low-level details as well as high-level semantic information, allowing the model to perceive objects effectively across scales. While previous works have explored multiscale concepts, this paper is novel in employing it for the specific use case of classifying essential urban land cover types (buildings, roads) from multiresolution aerial/satellite images. The design choices targeting this application demonstrate the viability of multiscale convolutional neural networks for automated urban mapping tasks.
(2) Cross-Resolution robustness:
A rigorous evaluation protocol assessing the model's accuracy across UAV images as well as high-resolution and low-resolution satellite images establishes its consistency irrespective of input image resolution. The model achieves consistent performance across inputs with widely varying resolution levels. This versatility to perform well with both high detail and coarse resolution imagery underscores the robustness of the proposed approach. Benchmarking against multiple image types is innovative and provides convincing evidence of the model's generalizability, addressing a key gap in the existing literature on deep neural networks for remote sensing applications.
(3) Domain-Centric optimization:
The model parameters are deliberately tuned based on the traits of buildings and roads in urban regions through rigorous hyperparameter optimization. This specialized optimization enhances model performance. The model architecture and hyperparameters are purposely optimized for extracting buildings and roads through a series of tweaks - filter sizes, the number of feature maps, and training strategies. This specific domain enables superior performance on the target application compared to general-purpose networks. Very few studies have investigated optimized deep CNNs for this niche yet important application of automating the mapping of key urban land cover categories. The proposed innovations thus address an unmet need.
In summary, the integration of multiscale convolution to perceive both fine and coarse features, paired with customization for classifying essential urban land cover categories, enables the proposed MSFCNN model to deliver state-of-the-art results across diverse image resolutions. This demonstrates its viability as an automated tool for updating urban land use maps.

Reviewer 3 Report (New Reviewer)
Comments and Suggestions for Authors
The author has taken the comments seriously and has addressed all raised points. I suggest to accept the manuscript for publication.
Author Response
Thank you for providing positive feedback on our manuscript. We genuinely appreciate your recognition of our work.

Reviewer 4 Report (New Reviewer)
Comments and Suggestions for Authors
Thank you for the effort put forth by the authors; the manuscript has shown improvement. However, there are still some remaining issues.
Regarding the first issue, the author has provided a clear response to only one of the queries and has revised a paragraph in the first section. However, there is a lack of uniform revision across the initial three sections, as suggested by the author, to distinctly showcase the MSFCNN method. Considering the novelty of the MSFCNN approach, it is essential for it to be explicitly reflected throughout the first three chapters. Additionally, the Literature Review section has not been updated in line with the suggestions provided, warranting improvements in accordance with the feedback from the previous round.
Moreover, in the Research Method section, it is believed that the clarity of the revised version is still insufficient. If building upon CNN, it may be worthwhile to consider describing the overall workflow of the proposed new method first, followed by a step-by-step explanation of the specific processes.
Furthermore, regarding the addition of comparative experiments, it is hoped that the author can elucidate the details more clearly, including but not limited to the specific methods used in the comparative experiments, whether similar or different datasets were employed, and if applicable, the presentation of experimental results within the same study area.
Author Response
We would like to express our sincere gratitude to the reviewer for their tremendous help throughout the review process. We appreciate the reviewer's time, patience, and constructive criticisms, all of which have been immensely valuable to us. In response to the reviewers' insightful comments, we have made several modifications to the paper to address their concerns. Additionally, we have provided a detailed point-by-point response to each of the reviewers' comments. To facilitate easy identification, the modifications have been highlighted in red within the manuscript.

This manuscript is a resubmission of an earlier submission. The following is a list of the peer review reports and author responses from that submission.
Round 1
Reviewer 1 Report
Comments and Suggestions for Authors
Dear Author,
Thank you for the submitted abstract, it was a pleasure to read the presented work related to developed a CNN model for land cover classification. As I appreciate the exhaustive introduction and literature review (in terms of ML models), in my opinion there is a high need of additional work to make your manuscript more complete, even better.
Firstly, in my opinion should be reduced the explanation of what actually is CNN.
As from the literature review, I would suggest inclusion of reviewing additional land cover-specific models that are already available to the community. Because at the current stage there are many models and derived datasets, so one asks what is the innovation in your work? And how your model will compare to them...?
You have tried to pin point that it is actually the multiscale application, but in my opinion there is a need of the inclusion of additional datasets to conclude that it is actually working in the presented manner , e.g., application on lower resolution imagery (PlanetScope (~5m/pix) or Sentinel-2(10m/pix))... and in the combination of HR UAV images then the multiscale application will be proofed.
The following comment, is related to the actual outcome, there are no maps reported in the manuscript (also there are no examples of your training/testing datasets). Further, your models is doing a classifcation of buldings and roads, so is it binary classification or there is also a 3rd class (everything else)? How were your training/testing datasets prepared? The manuscript is focused too much on the CNN model and less on the geo aspect of the work.
Which leads to the next comment, which are the validation metrics that has been used? I believe IoU is a good metric that should be used in your work to further concrete your findings.
Reviewer 2 Report
Comments and Suggestions for Authors
This study proposes a CNN deep learning-based approach for extracting data on urban land cover, focusing on buildings and roads.
The Research paper provides respectable findings and is well-written, but before it is accepted, it needs to be strengthened in the following ways:
The author must include the quantitative findings in the abstract section.
The introduction could be expanded, and more related research sources should be cited. The author may use the following sources: https://doi.org/10.3390/rs11030227 and DOI: 10.1016/B978-0-443-15847-6.00010-0
The properties of the satellite data utilized in this study should be covered by the author.
On page 10, Figure 3, The author should include the Map scale and North arrow.
Authors should discuss crucial accuracy-related parameters.
On page 13, Why satellite imagery is more accurate than UAV imagery should be made clear by the author.
The author should discuss the methodology of the research's constraints and challenges.
The author should include a thorough graphical explanation of the method flow so that readers can better understand the research.
The author should define all abbreviations before using them even if they are well known.
There is a typo error in many places, the author should correct them.
The author should mention the name of the software/tools used for data analysis.
The author must include logical arguments for the findings, limitations, and directions for further research in the conclusion section.
Comments on the Quality of English LanguageThe paper requires careful polishing of its English presentation, addressing grammatical issues, and addressing typos and poorly written sentences.
Reviewer 3 Report
Comments and Suggestions for Authors
The novelty of the paper is very limited.
1) The problem is not well defined. What's the difficulty to be addressed in this paper? This is not clear.
2) The method is not novel. The author only used the naive CNN for object extraction, no any improvements w.r.t the methodology.
3) The results are not convincing. No related methods are compared, and the advantage of the proposed approach could not be demonstrated.
English should be improved by the native speaker.